# Advances in 3D Bioprinting: Techniques, Applications, and Future Directions for Cardiac Tissue Engineering

**DOI:** 10.3390/bioengineering10070842

**Published:** 2023-07-16

**Authors:** Catherine A. Wu, Yuanjia Zhu, Y. Joseph Woo

**Affiliations:** 1Department of Cardiothoracic Surgery, Stanford University, Stanford, CA 94305, USA; catherinewu618@gmail.com (C.A.W.); yuanjiaz@stanford.edu (Y.Z.); 2Department of Bioengineering, Stanford University, Stanford, CA 94305, USA

**Keywords:** cardiac tissue engineering, bioprinting, biomaterials, bio-inks

## Abstract

Cardiovascular diseases are the leading cause of morbidity and mortality in the United States. Cardiac tissue engineering is a direction in regenerative medicine that aims to repair various heart defects with the long-term goal of artificially rebuilding a full-scale organ that matches its native structure and function. Three-dimensional (3D) bioprinting offers promising applications through its layer-by-layer biomaterial deposition using different techniques and bio-inks. In this review, we will introduce cardiac tissue engineering, 3D bioprinting processes, bioprinting techniques, bio-ink materials, areas of limitation, and the latest applications of this technology, alongside its future directions for further innovation.

## 1. Introduction

Cardiovascular diseases (CVDs) are pathologies affecting the myocardium, heart valves, or vasculature [1]. CVD is the leading cause of morbidity and mortality in the United States and is currently on the rise [1,2,3]. This is attributed to the heart being one of the least regenerative organs in the body due to cardiomyocytes’ limited renewal potential and a lack of endogenous cardiac stem cells [4,5,6]. Treatments for CVD include lifestyle modifications, therapeutic methods, or surgical interventions such as heart transplantation [2,6].

Tissue engineering has the potential to develop practical replacements for damaged tissue through material engineering, life sciences, and computer modeling [5,7]. Specifically, the field of cardiac tissue engineering aims to repair damaged or diseased blood vessels, heart valves, cardiac muscle, and other defects, with a long-term goal of artificially rebuilding a full-scale organ that matches native structure and function [1,6,8]. In the past several years, three-dimensional (3D) bioprinting has been progressive in its technical advancements and promising applications. 3D bioprinting involves the layer-by-layer deposition of biomaterials to fabricate architectures, including structural scaffolds, functional tissues, organ models, and more [5,7]. The four main bioprinting techniques are droplet-based, laser-assisted, stereolithography, and extrusion-based, and are used with a range of natural, synthetic, and hybrid bio-inks.

This review will first introduce cardiac tissue engineering, then highlight various aspects of 3D bioprinting, specifically the engineering process, bioprinting techniques, and bio-ink materials. Finally, we will summarize recent cardiac applications of 3D bioprinting, its limitations, and future directions for its advancement and integration in biomedical applications and regenerative medicine.

## 2. Cardiac Tissue Engineering

### 2.1. Cardiovascular Diseases

The heart has limited regeneration potential and cardiomyocytes are threatened by factors such as ischemia, necrosis, and apoptosis which lead to heart disease or failure [9]. Additionally, cardiomyocytes exhibit low cell turnover rates of 0.3–1% per year [10]. Following myocardial infarction, up to one billion cardiomyocytes are lost, forming scar tissue unable to transmit electrical signals and contractile activity [11]. Prolonged cardiac damage increases the risk of CVD, as well as irreversible acute and chronic heart failure [5].

Many therapies for CVD are restricted to symptomatic treatment, which lacks the ability of in situ cardiac resurrection [6]. Heart transplantation for end-stage heart failure is limited by a lack of organ donors, immune rejection [12], surgical complications, anticoagulant therapy [13], and restricted durability. Prosthetic valve devices pose a risk of thromboembolic complications and lifetime bleeding [14,15]. Specifically, xenografts of bovine or porcine heart valves are commonly associated with structural valve failure [16]. Autografts used in coronary artery bypass graft surgeries are limited by the availability of suitable conduits for patients, as well as additional risks associated with surgeries [17,18]. Limitations of current CVD therapies lead to a recent direction seeking regenerative cell sources as a bioengineering solution. Clinical studies have shown promising results of stem cell transplants, from the cellular level to engineered heart tissues [19,20,21]. However, direct injection of stem cells often results in low cell viability [5]. Using induced pluripotent stem cells (iPSCs) for in vitro and in vivo cardiac reprogramming is also restricted by a lack of mechanical support and directional guidance from the nearby extracellular matrix (ECM), resulting in cell lines that are not morphologically sufficient for higher-dimension organization necessary for anatomical shaping and organic integration [6].

### 2.2. Tissue Engineering

A regenerative engineering solution addressing the limitations of CVD therapies is the field of tissue engineering, a multidisciplinary approach that offers strategies to replace damaged tissue. Cardiac tissue engineering is motivated by the need for functional tissue equivalents in repairing heart defects and studying cardiac tissue. Approaches in tissue engineering are determined by cell source, biomaterials, oxygenation, media perfusion, and exposure to physiologically relevant stimuli [22,23,24]. One important aspect of tissue engineering is scaffold design. Scaffolds act as a supportive framework promoting cellular interactions, adhesion, proliferation, and differentiation, as well as providing support to the developing tissue [8]. Scaffolds can also function as a vehicle for delivering and incorporating growth factors for controlled and enhanced tissue growth [7]. Scaffold fabrication techniques include but are not limited to micropatterning [25], electrospinning [26], thermally induced phase separation [27], and hydrogel matrix systems [28]. An ideal scaffold mimics the natural ECM of the tissue’s implantation site for suitable cell development and regeneration [29].

### 2.3. 3D bioprinting

Three-dimensional (3D) bioprinting enables the fabrication of 3D architecture of complex spatial patterns through the layer-by-layer deposition of a range of biomaterials [5]. 3D bioprinting allows for control over construct fabrication and cell distribution, with a printing resolution close to the finest features of tissue microarchitecture from ten to a few hundred micrometers (μm) [30,31,32]. With substantial repeatability, reproducibility, controllability, and printing throughput, 3D bioprinting can produce customized devices with continuous and stable biological patterns. This technology offers potential for tissues, organs, prosthetics, drug delivery systems, and, ultimately, high-resolution simulations of the heart for innovative explorations of myocardial tissue repair and regeneration [5,6,33].

The bioprinter is encapsulated by a set of consecutive manufacturing operations guided by integrated computer numerical control machinery. Basic industry references are indicated by fundamental operating parameters, crosslinking, and print rheology measurements [34,35,36]. During the printing process, the platform’s movement is governed by coordinates saved in file format, such as a g-code, that can be easily followed by the printer [7].

Print conditions, such as printing nozzle aperture, printing speed, printing temperature, number of printed layers, and layer thickness, can vary widely. Each variable can greatly impact cell survival and construct fidelity [6,7,37,38]. Printability should be optimized to improve the fabrication process and construct properties [7].

Following construct completion, biological and mechanical assessments are performed for the physiological stability and functionality of the printed structures. Factors for consideration include structural fidelity, mechanical stability and elasticity, structure swelling and degradation, cell viability, and cell-material interactions [39]. For cardiac tissue bioprinting, construct evaluations include observing cardiac biomarkers, effective contractile forces, spontaneous action potential, overall calcium regulation, and more [6]. Post-fabrication steps, which are required to accustom the biomaterial to new functions, are selected depending on different properties of cardiac tissues, such as contraction [40], blood and nutrient perfusion [41,42,43], and electrical signaling [1,44].

## 3. Bioprinting Techniques

3D bioprinting can be classified into four main techniques, namely droplet-based, laser-assisted, stereolithography and digital light processing, and extrusion-based bioprinting. A summarization of these four main bioprinting techniques is presented in Table 1.

### 3.1. Droplet-Based Bioprinting

Droplet-based bioprinting involves replacing the printer’s ink cartridge with biological material for continuous droplets printed through an ejector [8,45], illustrated in Figure 1. Droplet-based bioprinting offers advantages such as its compatibility with a range of biomaterials with remarkable cell viability greater than 90% [8]. Micro-scale droplets that are 1–100 picolitres with densities of up to 10,000–30,000 cells per drop can be printed at a deposition rate of 1–1000 drops per second [46,47], which results in high construct resolution (<100 μm). Droplet-based bioprinting also allows for variable biomaterial concentration gradients with controllable cell growth factors by altering drop density and/or size [5].

Droplet-based bioprinting can be classified into electrohydrodynamic jet, acoustic, microvalve-based, and, most commonly, inkjet bioprinting [1]. Drop-on-demand inkjet bioprinting is based on an automated delivery of a controlled volume of bio-inks, usually containing cells, in a droplet fashion to predefined locations [5]. Materials’ deposition relies on persistent external mechanical force and gravity, which creates a 3D structure directed by an established route, as the base elevator [6] is electronically controlled for z-axis movement [8]. Drop-on-demand inkjet bioprinting can be further classified into continuous-inkjet bioprinting, electrohydrodynamic jet bioprinting, and drop-on-demand inkjet bioprinting based on varied droplet motivation mechanisms [7,48]. Though inkjet-based bioprinting of cardiac tissues is still in its infancy stage, many developments have been studied with this technique [49,50,51]. Biomaterials compatible with inkjet bioprinting usage include hydrogels, fibrin, agar, alginate, and collagen [52]. Specific mechanisms of drop-on-demand inkjet bioprinting, which offer negligible impact on cell viability, include thermo-based, piezoelectric-based, and electrostatic inkjet bioprinting [8,53]. Thermal-based bioprinting uses an ink chamber extruded through several small nozzles [1]. Localized heating produces short currents and pressure pulses to the heating element, raising the temperature of the ink’s surrounding element, which results in bubble formations that eject ink droplets [8]. However, thermal-based bioprinting is not commonly used in tissue engineering due to the loss of cell activity or hydrogel denaturation caused by high temperatures, which may be upwards of 200 °C [48,54]. Piezoelectric-based bioprinting uses piezoelectric crystals at the rear end of the bio-ink chamber that vibrate in response to electrical charges. These inward vibrations force small amounts of bio-ink through the nozzle [55]. Yet, the generated acoustic waves work in a frequency range of 15 to 25 kHz, which may cause cell damage [5,8]. Electrostatic inkjet bioprinting utilizes instantaneous increases in volume to achieve ejection. Applying impulse voltage to a platen and motor bends the platen for bio-ink extrusion [56].

There are several drawbacks of droplet-based bioprinting, such as the inability of the printer to extrude a continuous flow of bio-ink, which limits the mechanical and structural integrity of printed constructs [53,57]. To counteract shear stresses from crosslinking processes, bio-inks for droplet-based bioprinting must exhibit low cell densities compared with other printing techniques [58]. The dispensing mechanism and non-contact nature of the printer require bio-inks with lower viscosity (3.5–20 millipascal seconds, <10 cP), resulting in constructs lacking structural integrity and mechanical strength [59,60,61], as well as non-uniform droplet size, low droplet directionality, mechanical and shear stress to cells, and frequent nozzle clogging [5]. 

### 3.2. Laser-Assisted Bioprinting

Laser-assisted bioprinting relies on sensitive optical guidance, where a high-intensity laser propels bio-ink droplets in a non-contact mode [5,6], as shown in Figure 2. Laser-assisted bioprinting uses a pulsing laser beam and two parallel slides—a donor and collector—to produce the desired construct. A laser-absorbing metal beneath the donor slide is covered by the biomaterial to be transferred. As the laser pulses—with energies ranging from 65 nJ to 190 uJ—are absorbed by this metal, biomaterials from the donor slide fall through the evaporated metal onto the collector slide [62,63,64]. This technology allows for high-resolution deposition of biomaterials in the solid or liquid phase [7], creating 2D or 3D models through stacking of droplets [65]. Advantages of laser-assisted bioprinting include eliminated orifice nozzle clogging or contamination factors [1], high cell viability attributed to the low mechanical stress on cells during printing [5,65], a wide range of biomaterials usable [5], ability to deposit cells with high resolution, and high printing speeds [18,66]. With regard to cell deposition, laser-assisted bioprinting allows for control over the number of cells per droplet and high cell densities, where the printing resolution is dependent on parameters such as biomaterial viscosity, layer thickness, and laser influence [16,18]. On the other hand, laser-assisted bioprinting is time-consuming, costly, not commercially available, and only capable of producing small-sized structures, resulting in limited clinical applications [1,5].

### 3.3. Stereolithography and Digital Light Processing Bioprinting

Both stereolithography and digital light processing bioprinting are based on the polymerization of photo-cross-linkable materials (light-sensitive polymers) using a precisely controlled light source [5,7]. The respective schematics of stereolithography and digital light processing are illustrated in Figure 3.

Stereolithography involves a laser-assisted bioprinting system to produce structures by photocuring photopolymerizable liquid polymers, creating more realistic microstructures compared to other techniques [67]. Photocuring and photopolymerization occur as liquid polymers are crosslinked by exposing predesigned patterns using ultra-violet (UV), infrared, or visible light laser beams. Stereolithography operates using a UV light source, a liquid photopolymer resin tank, and a three-axis motion platform [68]. A platform is lowered into the resin tank, creating a thin layer of liquid between the platform and the bottom of the tank. The laser is guided through a window at the bottom of the tank and draws cross sections of the 3D construct while selectively polymerizing the biomaterial. Once one layer is finished, it is removed from the bottom of the tank, allowing fresh resin to flow beneath it. The platform is then lowered and the process is repeated [5]. Stereolithography offers high printing resolution of 200 nanometers [69], fast printing speed [67,70,71], high cell viability due to the nozzle-free mechanism [72], and construct accuracy [1]. A major drawback of stereolithography is that its UV light sources are expensive and affect cell viability [67,73]; this challenge is overcome with the usage of visible light stereolithography bioprinting [69,74].

While stereolithography uses a UV laser beam to solidify materials, digital light processing employs a digital light projector as a light source. A micromirror device with user-defined patterns consecutively loaded to turn on/off mirrors to reflect incoming UV light [75,76,77]. This process selectively solidifies photocurable bio-inks in a layer-by-layer process controlled by a moveable stage along the z-axis. As such, entire layers may be fabricated simultaneously in a single exposure phase for a reduced printing time, and the nozzle-free technology avoids clogging and excessive shear stress to cells [8].

Both stereolithography and digital light processing are limited by cytotoxic effects that may result from photo initiations and UV light exposure [78,79]. Additionally, the range of bio-inks used in stereolithography and digital light processing bioprinting is limited by their need to be readily crosslinked through light irradiation in order for the uncured bio-ink to interface with the cured layers [80].

### 3.4. Extrusion-Based Bioprinting

Extrusion-based bioprinting utilizes a computer-controlled system to continuously extrude bio-inks, specifically viscoelastic biomaterials in filaments [81], using layer-by-layer extrusion with the nozzle free to move in the x-y-z directions and an adjustable printer stage to fabricate a 3D construct [1,5]. Among extrusion-based bioprinting approaches, presented in Figure 4, the pneumatic-based technique uses pressured air at a controlled volume flow rate to drive fluid dispensing systems’ constant extrusion of bio-ink, whereas the piston/screw-based technique mechanically forces biomaterials out of the nozzle [1,8,82]. Regarding printability, bio-inks with viscosities in the range of 30 to 6 × 10^7^ mPas have been reported to be printable [83]. Factors impacting printability include viscosity adjustability, bio-ink phase (e.g., liquid phase) prior to extrusion, and biomaterial-specific parameter ranges [84]. Advantages of extrusion-based bioprinting include the scaffold’s tunable biodegradability properties, which can match ECM regenerate rate [85], simultaneous usage of multiple biomaterials and/or varying cell types with multi-nozzle bioprinters, deposited cell densities close to physiological (cardiomyocyte) densities, and convenience and affordability [8]. Limitations include its low printing resolutions and difficulty in obtaining precise cell patterning and organization [5].

The two main extrusion methods of extrusion-based bioprinting are direct and indirect methods. Direct extrusion-based bioprinting is based on the extrusion of bio-inks into a cell-friendly environment. Post-extrusion, hydrogels solidify to form the 3D construct, and cells proliferate to undergo tissue remodeling [5]. Indirect extrusion-based bioprinting uses fugitive ink that is removed by a thermally induced de-crosslinking process, leaving only the hollow structure’s vascular network [25]. A recent extrusion-based method employs a core–shell nozzle and crosslinking agent printed simultaneously with the bio-ink extrusion [86,87].

Extrusion-based bioprinting has been employed in many applications [88,89,90,91,92,93,94,95,96]. For example, in coaxial nozzle-assisted bioprinting, encapsulated cells are extruded through a central needle while the crosslinking solution remains in the needle’s outer portion during the printing process [93,97,98,99,100,101,102,103,104]. Fused deposition modeling involves the layer-by-layer extrusion and fabrication of polymeric thermoplastic materials through a heated nozzle, allowing for highly customizable morphologies and tunable mechanical properties [105,106]. Scaffold-free applications [107,108,109,110], arising from scaffolds’ high probabilities of rapid degeneration [92,111,112], enable the direct printing of living cells into a predefined pattern, where an inflammatory response due to scaffolding is avoided, and cells can immediately differentiate in the 3D environment [86].

Freeform Reversible Embedding of Suspended Hydrogels (FRESH) is another rising technique where hydrogel bio-inks are extruded into another hydrogel support medium (FRESH) [113,114,115,116]. This method offers potential for fabricating complex structures, high construct fidelity, tunable mechanical properties to create a suture-able tissue, reinforcing cell survival through indirect extrusion-based bioprinting, reduced influence bio-inks’ rheological properties attributed to FRESH’s compatibility with lower viscosity bio-inks, and low costs [5]. However, drawbacks include print repeatability and precision [5], structure integrity and cell viability jeopardized by mechanical forces required for FRESH removal [8], and the impacts of FRESH support bath conditions, which have been previously studied in our work [117].

## 4. Bio-Inks and Biomaterials

Bio-inks used in 3D bioprinting are biomaterial solutions in hydrogel form, often containing or encapsulating target cell types and growth factors, extruded for construct fabrication [118]. Ideal bio-inks are non-toxic, non-immunogenic, and offer mechanical stability and integrity. Demonstrating appropriate biodegradability rates and promoting cell adhesion may also be beneficial depending on the specific project needs. Other factors for consideration include cytological elements [6], gelatin properties and crosslinking ability, cost, print time, industry scalability, and permeability [119]. Specifically, crosslinked hydrogels are highly porous, supporting tissue reconstruction and regeneration by allowing cell–cell adhesion, proliferation, differentiation, and migration to populate scaffolds and nutrient delivery for cells’ metabolic needs [1,9]. Additionally, bio-inks may also contain additives to improve conductive properties, which is necessary for recreating the native cell environment [5]. Bio-inks for bioprinting in cardiac tissue engineering should be selected depending on the specific application. Of many features, bio-inks used in cardiac tissue engineering should shield cardiac cells against varying pressure levels and shear stresses [8], as well as assisting in the formation of vascular supportive substructures for blood vessels on the micro-level. These factors are important for biomaterials to support tissue reconstruction and regeneration, as well as to culture healthy tissues [9]. Maintaining appropriate rheological abilities is also necessary for balancing the internal and external shear forces on the construct during bioprinting [120].

Controlling cell microenvironments for tissue-engineered scaffolds is important in directing cell behavior within scaffolds both spatially and temporally. This can be achieved by inducing and maintaining cell alignment, which plays a role in cell behavior and tissue functionality. Specific to cardiac tissue engineering, cardiac scaffolding tissues should have microenvironment and contraction properties of cardiomyocytes. Biomaterials used as scaffolds must form a biomimetic ECM to promote cell adhesion and differentiation and 3D organotypic cultures. A review of some of the common biomaterials used in cardiovascular bioprinting is shown in Table 2. Tissues composed of ideal biomaterial elastomers for cardiac tissue engineering, such as exhibiting a low Young’s modulus, having high elongation, tensile strength, elasticity, and tunability, as well as demonstrating stable degradation characteristics, are integral for functionality [9,121].

Natural biomaterials are derived from the organ of interest as decellularized extracellular matrices (dECM), with high biocompatibility and intrinsic bioactivity mimicking the native ECM [5,8]. These biomaterials may be derived from polysaccharides (e.g., agarose, alginate [122,123,124,125,126], chitosan), proteins (e.g., collagen [126,127,128,129,130,131,132,133], gelatin [134], fibrin [135]), glycosaminoglycans (e.g., hyaluronic acid [136,137], heparin), keratin, Matrigel and dECM [138], silk fibers [139], and more. However, natural biomaterials have limitations as bio-inks due to low mechanical strength [6], insufficient mechanical properties, variability, immunogenicity, and low tunability [5].

On the other hand, synthetic biomaterials span polyacrylic derivatives, polycaprolactone [5], polyethylene glycol copolymers, polyglycolic acid [131], polylactic acids [140], poly(DL) glycolate, polyphosphazene, and synthetic peptides, polyvinyl alcohol and derivatives, Pluronic [141], and more [142]. Synthetic biomaterials are compatible with a wide range of physical and chemical modifications [5] and offer better physical integrity, higher mechanical strength, enhanced control of printability, low immunogenicity, and no batch-shift variability [6,8]. Their durable framework and biocompatibility also aid in the formation of grafts for cardiac implants and bypass surgeries [1]. Yet, drawbacks include inferior biocompatibility compared to natural biomaterials [6], brittleness, lack of flexibility and elasticity, and difficulty in mimicking tissue softness, stretchability, and electability, such as in blood vessels and heart muscles [8].

Hybrid biomaterials demonstrate the cell-supportive properties of natural polymers in conjunction with the mechanical properties and tunability of synthetic polymers. Examples include blending alginate and gelatin methacrylate [143,144], as well as alginate and polyvinyl alcohol bio-inks [145]. Multi-material constructs offer benefits such as improving structural complexity, adjustable growth factors, loading different cell types in different zones to mimic natural cellular diversity and activity, and the simultaneous deposition of biomaterials with varying physical and chemical properties—useful in fabricating tissues from varying regions with varying properties [7]. 

## 5. Bioprinting Applications in Cardiac Tissue Engineering

### 5.1. Cellular Sources

Cellular sources for cardiac tissue engineering bioprinted constructs should embody fast cell proliferation, easy differentiation and maturation into the target cell type(s), easy accessibility to cell sources of autologous origin, and non-antigenicity with immunity to pathogens [146]. Ideal candidates for cardiac tissues include native cardiomyocytes, progenitor cells, and stem cells, such as bone-marrow- or cord-derived mesenchymal stem cells [147] and cardiac stem cells [148]. 

Stem cells offer flexible degrees of self-renewal, differentiation ability into cardiomyocytes, and high proliferation [5], as well as the potential to reduce immune rejection of grafts, decrease thrombogenic effects, and availability on-demand [149]. Disadvantages include their ethical controversies, immature phenotypes, absence of transverse tubules, reduced contractility, and altered metabolic and electrophysiological properties in comparison with adult cardiomyocytes [5]. Regarding stem cell usage in 3D bioprinting, one study employing gelatin-fibrin bio-ink explored the ratio of co-culturing varying types of cardiac cells such as cardiomyocytes, cardiac fibroblasts, endothelial cells, and demonstrated hetero-cellular coupling of different cell types via bioprinting [112]. Relatedly, one study showed hetero-cellular crosstalk between two dissimilar cell types (C2C12 myocytes and STO fibroblasts) at the interface of printed cell sheets in the multi-layered tissue [94]. The cellular crosstalk observed in these studies demonstrated applicability for future tissue engineering of complex tissues.

### 5.2. Cardiac Tissue and Patches

Cardiac patches have properties desirable for clinically relevant cardiac repair [150]. They have been specifically used to repair or replace diseased cardiac tissues and restore cardiac functionality to an extent [1]. Recent advances have been made to fabricate tissues using 3D bioprinting technology. In 2009, Cui et al. studied the formation of microvascularization through simultaneous printing with human vascular endothelial cells and fibrin scaffolding [60]. The group found the 3D tubular structure in printed patterns, with cell alignment present inside channels and proliferation forming confluent linings. This study showed the implications of simultaneous cell and scaffold printing in cell proliferation and microvasculature. Later in 2013, Shin et al. seeded neonatal rat cardiomyocytes onto carbon nanotubes that incorporated photo-cross-linkable gelatin methacrylate hydrogel to create cardiac patches [151]. Excellent mechanical integrity and advanced electrophysiological functions were observed in the constructs, demonstrating the potential of carbon nanotube incorporation in fabricating multifunctional cardiac scaffolds. A more recent study by Jang et al. in 2017 used stem cell-laden dECM bio-inks to print pre-vascularized and functional multi-material 3D structures [89]. Once developed, the stem cell patch was shown to promote strong vascularization and tissue matrix formation when implanted in hearts in vivo. The patch showed reduced cardiac hypertrophy and fibrosis, increased cell migration from the cardiac patch to the myocardial infarct site, formation of neo-muscle and capillaries, and overall improved cardiac functions. Most recently, in 2019, an approach by Noor et al. involved printing thick, vascularized, and perfuse-able cardiac patches matching patients’ immunological, cellular, biochemical, and anatomical properties [152]. Patients’ omental tissue was reprogrammed into iPSCs and then differentiated into cardiomyocytes and endothelial cells, where they were combined with hydrogels as bio-ink in fabricating cardiac tissue and blood vessels. Noor et al. demonstrated the ability to print vascularized patches according to patient anatomy and improved blood vessel architecture. Cellularized human hearts with natural architecture were also printed in this study, indicating a potential for engineering personalized tissues and organs.

The in vitro fabrication of cardiac tissue is more sophisticated than bioprinting cardiac patches and requires ordered arrangement of multiple cell types for a multi-scale vasculature network, lymphatic vessels, and neural and muscle tissues [1]. Tissues must also possess electrical pacing for autonomous contractions. Many recent studies have demonstrated promising results of bioprinting cardiac tissues.

For example, over a decade ago, Xu et al. reported fabricating bioengineered cardiac pseudo tissues—specifically contractile cardiac hybrids and a “half-heart” structure—with beating cell responses using an inkjet-based bio-prototyping method [153]. In another study, Wang et al. used cardiomyocyte-laden hydrogel to bioprint cardiac tissue constructs with spontaneous synchronized contractions in vitro [154]. Progressive tissue development and maturation were shown, as well as physiologic responses to cardiac drugs, implying cardiac tissue engineering and pharmaceutical applications. Later in 2018, Maiullari et al. presented work with heterogeneous multi-cellular constructors of human umbilical vein endothelial cells and iPSC-cardiomyocytes [100]. After encapsulating cells in hydrogel strands containing alginate and polyethylene glycol-fibrinogen, extrusion through microfluidic bioprinting technology fabricated a cardiac tissue consisting of iPSC-cardiomyocytes. The human umbilical vein endothelial cells provided the printed tissue with different defined and blood vessel-like geometries, which aided tissue integration with host vasculature. As shown, 3D bioprinting technologies have offered relevant implications in the generation of cardiac tissues with properties comparable to the native tissue environment. An overview of recent bioprinted cardiac constructs is summarized in Table 3.

Overall, tissue-engineered cardiac muscles may facilitate research on the heart’s physiology and grant high-throughput drug screening platforms in vivo [8]. Advancements in cardiac tissue bioprinting can ultimately lead to enhanced performance and functionality of cardiac tissue constructs [155]. However, much research around printing limitations is needed before further clinical applications.

### 5.3. Full Heart Organoid and Organ

Engineering functional heart organs comparable to native anatomies is the long-term goal of cardiac tissue engineering [8]. An illustration of bioprinting and translation is shown in Figure 5. Significant progress has been made over the past few years in this field. In 2019, Noor et al. demonstrated the first use of fully personalized, non-supplemented bio-ink materials for printing hearts with mechanical properties closely resembling the properties of decellularized rat hearts [152]. Later that year, Lee et al. successfully printed five components of the human heart, including a tri-leaflet heart valve, neonatal-scale collagen heart, and human cardiac ventricle model using the FRESH bioprinting technique [156]. The bioprinted hearts accurately reproduced patient-specific anatomical structures with high resolution. With regard to bioprinted cardiac organoids, in 2020, Kupfer et al. generated macroscale tissue with geometric structures relevant to the cardiac muscle’s pump function. The human-chambered muscle pumps exhibited beating and continuous action potential propagation in response to cardiac drugs and pacing. Kupfer’s work has implications for the fabrication of organoids of this nature, with applications for cardiac medical devices and tissue grafting [157].

**Table 3 bioengineering-10-00842-t003:** An overview of recent bioprinted constructs in cardiac tissue engineering.

	Construct	Morphogenesis	Physiology
Xu et al., 2009 [155]	Cardiac hybrid “pseudo” tissue and “half heart”	Feline cardiomyocytes with alginate hydrogel	Microscopic and macroscopic contractile functions, excitation-contraction coupling, beating upon simulation, cardiomyocyte alignment, and attachment to alginate/laminin channels
Jang et al., 2017 [90]	Cardiac patch	Human dECM, c-kit+cardiac progenitor cells, mesenchymal stem cells.	Enhanced cardiac functions, reduced cardiac hypertrophy and fibrosis, increased cell migration from patch to infarct area, neo-muscle, and capillary formation
Wang et al., 2018 [156]	Cardiac tissue	Infant rat primary cardiomyocytes with fibrin-based composite hydrogel	Spontaneous synchronous contraction in culture, progressive cardiac tissue development, physiologic responses to cardiac drugs
Maiullari et al., 2018 [101]	Cardiac tissue	human umbilical vein endothelial cells and iPSC-cardiomyocytes with alginate and PEG-Fibrinogen hydrogel	High orientation index (different defined geometries, blood vessel-like shapes), function infiltration and integration of vasculature into constructs, development of large endothelial-like structures
Noor et al., 2019 [154]	Cardiac patch and cellularized hearts	Cells from human omental tissues reprogrammed/differentiated into cardiomyocytes and endothelial cells, ECM processed into hydrogel	Patient-specific functional vascularized patches, improved blood vessel architecture, elongated cardiomyocytes with massive actinin striation, anatomical structure, patient-specific biochemical microenvironment
Lee et al., 2019 [158]	Heart components (microstructure, vessels, ventricles, valves, neonatal scale human heart)	iPSC-cardiomyocytes and collagen	20 μm filament resolution, rapid cellular infiltration and microvascularization (microstructure) and mechanical strength (valve); synchronized contractions, directional action potential propagation, wall thickening (ventricles); patient-specific anatomical structure (heart).
Kupfer et al., 2020 [159]	Chambered organoid	iPSCs and ECM-based bio-ink subsequently differentiated into cardiomyocytes	Human chambered muscle pumps beat synchronously, built pressure, and moved fluid similar to a native pump; connected chambers enabled perfusion and replication of heart pressure/volume relationships

Despite these major advances, bioprinting a fully functional and comprehensively structured human heart has yet to be accomplished. For one, the heart’s intricate structure and anatomy require long print times and complex material selection, such as scaffold material and cell source. For example, the FRESH approach by Lee et al. used collagen for printing the heart model, but printing with cells requires further research for manufacturability and potential clinical translation [156]. Printing resolution is also a major challenge to overcome, as the average resolution of 3D bioprinted constructs ranges from tens to hundreds μm, but the native tissue anatomy requires a resolution of 5–10 μm at the minimum [8,51]. Another challenge is the proper environment for cell culture, which is necessary for cell differentiation, tissue maturation, functional vessel network integration, and overall mechanical stability.

Further improvement in mimicking the native cell environment is also required for personalized applications in clinical research. Bioprinting a full heart requires the development of innovative biomaterials with physiologic mechanical properties, high biocompatibility, and dynamic behaviors to sustain print architectures, cell viability, and promoted vascular innervation [5]. Biomaterials used for bioprinting must also match the physical, chemical, and biological properties of patient tissue for clinical translation [158]. To alleviate construct immunogenicity, biomaterials, including gelatin and gelatin methacryloyl, dECM, and polyethylene glycol, can be utilized to improve biocompatibility [159]. Additionally, the inclusion of conductive polymers, sacrificial hydrogels, or adjunctive anti-inflammatory compounds during printing may also be beneficial in integrating bioprinted tissues with the host [160]. Further development is also needed of the heart’s complex components of multiple cell types, ECM, and multi-scale blood pumping structures [8]. More specifically, one major challenge that needs to be addressed prior to clinical translation is the engineering of vascular networks for functional tissue. Vascular networks enable nutrient delivery to the tissue and proper tissue formation. However, bioprinting vasculature is currently limited by printing resolution and speed, which are necessary for accurate construct structures and cell viability [68]. To improve the regeneration of constructs for implantation, medical imaging technologies can be employed to customize implants to match patients’ defect site dimensions and shape [161]. 

Following successful printing, an appropriate ex vivo system may be needed to assess the heart’s valvular tissues for functionality and to condition and train the entire organ to tolerate physiologic pressure and volume. Additionally, constructs’ structural and mechanical integrity, as well as long-term functionality, are required for the application of bioprinted tissues in clinical settings [162]. On this note, as the cost of bioprinters falls rapidly, 3D bioprinted patient-specific multi-cellular tissues present a cost-effective therapeutic for long-term treatment [160].

Regulatory standards and approval are also required for bioprinted tissues for clinical use. These approval pathways are determined by construct quality, safety, and efficacy in both nonclinical and clinical studies [163]. Generally, tissue-engineered products used for medicinal purposes demand unique approval processes for healthcare applications and commercialization. In the United States, bioprinted medical devices are subject to control requirements from the Food and Drug Administration, the Food, Drug, and Cosmetic Act, and additional market approval from regulatory agencies. In the European Union, tissue-engineered product-based therapies and treatments follow regulations by the European Medicines Agency. This agency regulates manufactured products for clinical trials and evaluates the quality, safety, and efficacy of new treatments prior to approval for marketing. 3D-printed medical device products are in accordance with legislation such as the Active Implantable Medical Device Directive, Medical Device Directive, and In Vitro Diagnostic Medical Device Directive. In China, tissue-engineered medical products are regulated according to the Medical Devices Classification Rule by the China Food and Drug Administration, with a focus on final product utility. In Japan, tissue-engineered products for clinical practice are regulated by the Pharmaceuticals and Medical Devices Agency. In Korea, 3D bioprinted scaffold implants must comply with the Korean Good Manufacturing Practice standards. Additionally, in India, the Central Drug Standards Control Organization regulates tissue-engineered products as therapeutic drugs, as well as assisting in manufacture, import, and marketing. On an international scale, guidelines and standards for tissue-engineered medical products are presented by the International Organization of Standards and the American Society for Testing and Materials [164]. The clinical translation of bioprinted tissues concerns various ethical scrutiny, including cell source, processing procedures, cost, variability post-implantation, and construct ownership [163]. To overcome regulatory requirements, multidisciplinary research should be conducted by experts in biology and medicine alongside experts in additive manufacturing and material science to achieve optimization and scalability of bioprinted products [165].

## 6. Conclusions

Overall, 3D bioprinting employs various bioprinting techniques using different bio-ink compositions to create functional models and tissue constructs. The droplet-based, laser-assisted, stereolithography and digital light processing, and extrusion-based bioprinting methods all demonstrate various advantages and drawbacks ranging from nozzle extrusion to print speed and construct resolution. Natural, synthetic, and hybrid biomaterials have been employed and researched for cell-supportive properties and mechanical tunability. Recent applications in cardiac tissue engineering ranging from tissue models to whole organs are relevant to understanding regenerative medicine and clinical studies. 3D bioprinting offers important implications in cardiac tissue engineering, with the technology’s limitations requiring much further work to address. Overall, cardiac tissue engineering has significant implications in the field of clinical medicine for the repairment, regeneration, and replacement of injured heart tissue and the organ as a whole. Further innovation to address current limitations offers potential for promising applications in regenerative medicine and clinical translation.

## Figures and Tables

**Figure 1 bioengineering-10-00842-f001:**
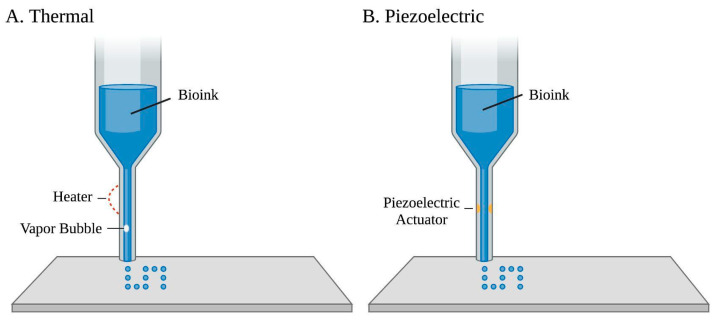
Two common droplet-based bioprinting approaches of a bio-ink printed through an ejector. (**A**) Thermal-based bioprinting uses localized heating to eject ink droplets along with vapor bubble formation. (**B**) Piezoelectric-based bioprinting involves an electric current that passes through a piezoelectric actuator to generate droplets.

**Figure 2 bioengineering-10-00842-f002:**
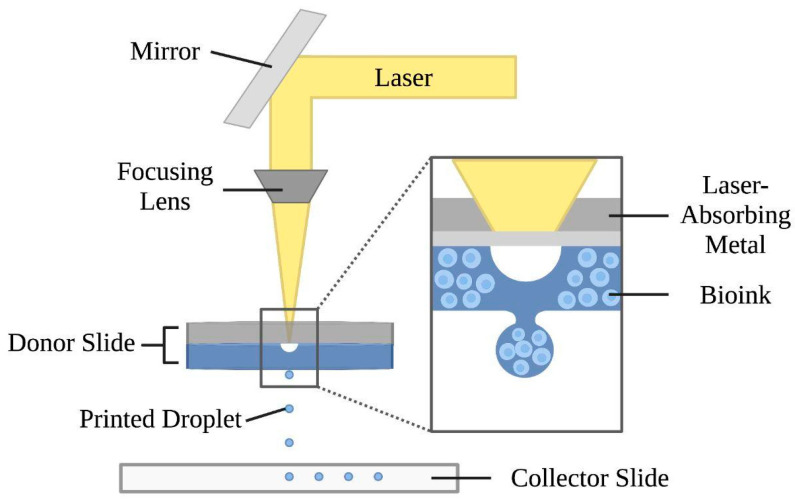
Mechanism of the laser-assisted bioprinting technique. This approach consists of a laser beam, a focusing lens, and two slides. The donor slide is composed of a laser-absorbing metal layer and a bio-ink layer. Laser pulses vaporize the metal layer and form droplets, which are ejected onto the collector slide below.

**Figure 3 bioengineering-10-00842-f003:**
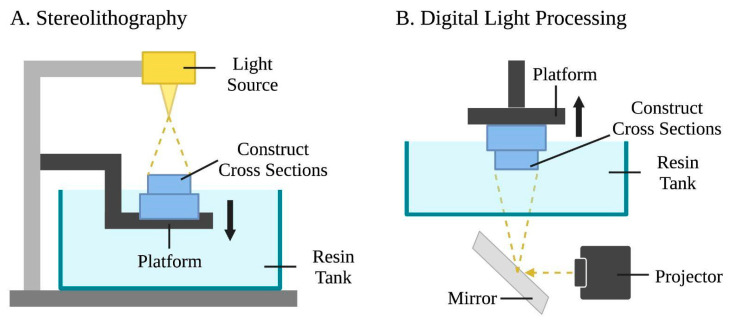
Schematic representations of stereolithography and digital light processing bioprinting. (**A**) Stereolithography uses a light source, a photopolymer resin tank, and a motion platform. The laser draws cross sections of the construct, and then the platform is lowered following the completion of each layer to let fresh resin flow beneath. This process is repeated layer by layer. (**B**) Digital light processing uses a light projector and mirror device that reflects the incoming light. Entire layers of the bio-ink are selectively solidified simultaneously while the platform moves vertically.

**Figure 4 bioengineering-10-00842-f004:**
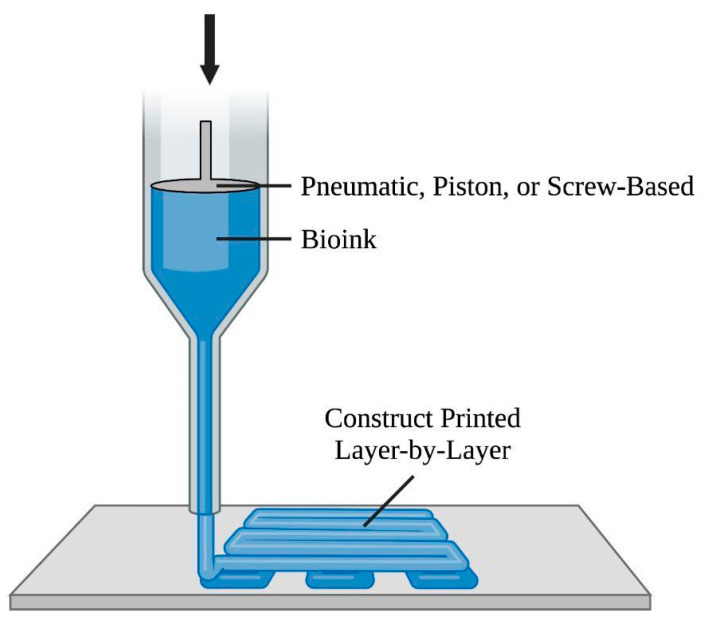
Diagram of the extrusion-based bioprinting approach. This method involves a computer-controlled system to direct the nozzle in the x-y-z directions. The pneumatic-, piston-, and screw-based techniques drive bio-ink, in filaments, out of the nozzle to form the construct.

**Figure 5 bioengineering-10-00842-f005:**
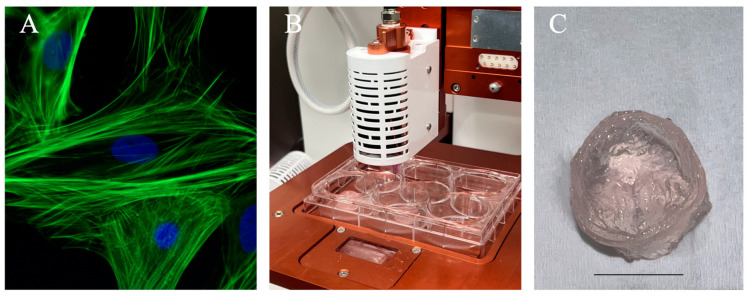
Illustration of a standard 3D bioprinting approach for generating an anatomical structure. (**A**) Immunofluorescence images of cardiomyocytes used in cardiac tissue engineering applications. Pictures here demonstrated neonatal cardiomyocytes stained for F-actin filaments (green) and nuclei (blue). Reprinted from Wikipedia Commons (2015). (**B**) Allevi 3 extrusion-based bioprinter setup to bioprint constructs using freeform reversible embedding of suspended hydrogel as a thermo-reversible support bath. (**C**) Trileaflet valve construct printed using 5% sodium alginate shown post-crosslinking. Scale bar = 10 mm.

**Table 1 bioengineering-10-00842-t001:** A summarization of four main bioprinting techniques.

	Droplet-Based	Laser-Assisted	Stereolithography & Digital Light Processing	Extrusion-Based
Advantages	Precise deposition, high cell viability, biomaterial compatibility, variable biomaterial concentrations, controllable growth factors	Non-contact printing, biomaterial compatibility, high cell viability, high cell densities	Non-contact printing, high resolution, high printing speed, cell viability, high cell densities	Biodegradability properties, simultaneous usage of multiple biomaterials, multiple nozzles, high cell densities, high viscosity bio-ink
Limitations	Inability to extrude continuous flow of bio-ink, low cell densities, low viscosity bio-ink	Time-consuming, high cost, limited construct size	Damage from UV exposure, cytotoxic effects, limited range of bio-inks	Low resolution, low precision

**Table 2 bioengineering-10-00842-t002:** A brief review of common biomaterials used in cardiac tissue engineering.

	Natural	Synthetic	Hybrid
Origin	dECM, polysaccharides, proteins, glycosaminoglycans, keratin, Matrigel	Polyacrylic derivatives, polycaprolactone, polyethylene glycol copolymers, polyglycolic acid, polylactic acids, poly(DL) glycolate, polyphosphazene, and synthetic peptides, polyvinyl alcohol and derivatives, Pluronic	Blending natural and synthetic polymers (e.g., alginate and gelatin methacrylate, alginate and polyvinyl alcohol bio-inks, etc.)
Characteristics	High biocompatibility, bioactivity, low mechanical strength, properties, tunability	Physical/chemical modifications, high mechanical strength, control of printability, low immunogenicity, low biocompatibility, lack of flexibility and elasticity	Improved structural complexity, adjustable growth factors, loading different cell types in different zones, simultaneous deposition of biomaterials with varying properties

## Data Availability

Not applicable.

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
