# Peer review of "Advances in 3D Bioprinting: Techniques, Applications, and Future Directions for Cardiac Tissue Engineering"

_bioengineering, 2023, doi:10.3390/bioengineering10070842_

Round 1
Reviewer 1 Report
Found this to be a well organized and interesting review of the topic. I think the paper is well put together.
1- The main question was addressed by the review
2- I believe that the topic is timely and addresses a specific gap in the literature
3- It is a concise and complete summary of a field that is not reviewed in the standard literature
4- I really cannot recommend any changes I would make
5-The conclusions are consistent
6- The references are appropriate
7- Everything else is acceptable
Reviewer 2 Report
The authors presented a review article called “Advances in 3D Bioprinting: Techniques, Applications, and Fu-2 ture Directions for Cardiac Tissue Engineering”. The review covers comprehensive aspects of the current development of 3D bioprinting for Cardiac Tissue Engineering. I have minor comments regarding the current structure, but I do have a general suggestion.
1. While several review articles have extensively covered the various techniques utilized in 3D bioprinting for cardiac tissue engineering (e.g., PMID:34335973), such as droplet-based, laser-based, stereolithography, extrusion-based methods, and the selection of biomaterials for bioink formulation, it would be beneficial for the authors to emphasize the progress made specifically in the application of 3D bioprinting for cardiac tissue engineering. Focusing on recent advancements and breakthroughs in the application will contribute to a more in-depth understanding of the current state-of-the-art in cardiac tissue engineering using 3D bioprinting technology.
2. In addition to discussing future perspectives and challenges, it is essential to include a thorough discussion on the clinical translation of bioprinting techniques for cardiac tissue engineering. Addressing the efforts required to accelerate the application of bioprinting in clinical settings would greatly enhance the review article. This discussion should encompass regulatory considerations, such as navigating approval processes and compliance with quality standards, as well as the development of robust and reproducible bioprinting protocols suitable for clinical use. Furthermore, exploring strategies to overcome challenges related to scalability, long-term functionality, immunogenicity, and integration of bioprinted cardiac constructs within the host tissue would be crucial in fostering the clinical translation of bioprinting technologies for cardiac tissue engineering.

Reviewer 3 Report
The article is well written, and I recommend it for acceptance with minor corrections and input.
